# Online List Labeling with Predictions

**Samuel McCauley**
Department of Computer Science
Williams College
Williamstown, MA 01267
`sam@cs.williams.edu`

**Benjamin Moseley**
Tepper School of Business
Carnegie Mellon University
Pittsburgh, PA 15213
`moseleyb@andrew.cmu.edu`

**Aidin Niaparast**
Tepper School of Business
Carnegie Mellon University
Pittsburgh, PA 15213
`aniapara@andrew.cmu.edu`

**Shikha Singh**
Department of Computer Science
Williams College
Williamstown, MA 01267
`shikha@cs.williams.edu`

## Abstract

A growing line of work shows how learned predictions can be used to break through worst-cast barriers to improve the running time of an algorithm. However, incorporating predictions into data structures with strong theoretical guarantees remains underdeveloped. This paper takes a step in this direction by showing that predictions can be leveraged in the fundamental *online list labeling problem*. In the problem, $n$ items arrive over time and must be stored in *sorted order* in an array of size $\Theta(n)$. The array slot of an element is its *label* and the goal is to maintain sorted order while minimizing the total number of elements moved (i.e., relabeled). We design a new list labeling data structure and bound its performance in two models. In the worst-case learning-augmented model, we give guarantees in terms of the error in the predictions. Our data structure provides strong theoretical guarantees— it is optimal for *any prediction error* and guarantees the best-known worst-case bound even when the predictions are entirely erroneous. We also consider a stochastic error model and bound the performance in terms of the expectation and variance of the error. Finally, the theoretical results are demonstrated empirically. In particular, we show that our data structure performs well on numerous real datasets, including temporal datasets where predictions are constructed from elements that arrived in the past (as is typically done in a practical use case).

## 1 Introduction

A burgeoning recent line of work has focused on coupling machine learning with discrete optimization algorithms. The area is known as *algorithms with predictions*, or alternatively, *learning augmented algorithms* [41, 44]. This area has developed a framework for beyond-worst-case analysis that is generally applicable. In this framework an algorithm is given a prediction that can be erroneous. The algorithm can use the prediction to tailor itself to the given problem instance and the performance is bounded in terms of the error in the prediction.

Much prior work has focused on using this framework in the online setting where learned predictions are used to cope with uncertainty [36]. This framework has been further used for warm-starting offline algorithms to improve the beyond worst-case running times of combinatorial optimization problems. This includes results on weighted bipartite matching [21], maximum flows [15], shortest-paths [13] and convex optimization [46].

37th Conference on Neural Information Processing Systems (NeurIPS 2023).

A key question is to develop a theoretical understanding of how to improve the performance of data structures using learning. Kraska et al. [35] jump-started this area by using learning to improve the performance of indexing problems. Follow up work on learned data structures (e.g. [38, 47]) have demonstrated their advantage over traditional worst-case variants both empirically and through improved bounds when the learned input follows a particular distribution; see Section 1.2.

The theoretical foundation of using learned predictions in data structures remains underdeveloped. In particular, while there has been some initial work analyzing data structures with predictions (i.e. [12, 40]) there are no known analysis frameworks showing how the performance of learned data structures degrade as a function of the error in prediction (as in i.e. [41]). In this paper, we take a step in this direction by showing how to leverage the learning-augmented algorithms framework for the fundamental **online list labeling problem**. We bound its performance in terms of the error in the prediction and show that this performance is the best possible for any error.

In the online list labeling problem[1], the goal is to efficiently store a list of *dynamic* elements in *sorted order*, which is a common primitive in many database systems. More formally, a set of $n$ elements arrive over time and must be stored in sorted order in an array of size $cn$, where $c > 1$ is a constant. The array slot of an element (i.e. its position in the array) is referred to as its *label*. When a new element arrives, it must be stored in the array between its predecessor and successor which may necessitate shifting (relabeling) other elements to make room. The goal of a list labeling data structure is to minimize the number of elements moved; we refer to the number of element movements as the **cost**. The challenge is that the insertion order is online and adversarial, and the space is limited. A greedy approach, for example, might end up moving $\Omega(n)$ elements each insert.

List labeling algorithms have been studied extensively for over forty years, sometimes under different names such as sparse tables [30], packed memory arrays [4, 7], sequential file maintenance [51–54], and order maintenance [19]. We call any data structure that solves the list labeling problem a **list labeling array (LLA)**. Itai et al. [30] gave the first LLA which moves $O(\log^2 n)$ elements on average over all inserts. This has since been extended to guarantee $O(\log^2 n)$ elements are moved on any insert [52]. These results are tight as there is an $\Omega(\log^2 n)$ lower bound on the amortized number of movements needed for any deterministic algorithm [10]. A recent breakthrough result has shown that a randomized LLA can achieve an improved bound of $O(\log^{3/2} n)$ amortized movements [3].

The LLA is a basic building block in many applications, such as cache-oblivious data structures [6], database indices [43] and joins [31], dynamic graph storage [18, 42, 50], and neighbor search in particle simulations [23]. Recently, LLAs have also been used in the design of dynamic learned indices [20], which we describe in more detail in Section 1.2.

Due to the their wide use, several prior works have attempted to improve LLAs using predictions heuristically. Bender et al. [7] introduced the **adaptive packed memory array (APMA)** which tries to adapt to non-worst case input sequences. The APMA uses a heuristic predictor to predict future inserts. They show guarantees only for specific input instances, and leave open the question of improving performance on more general inputs. Follow up data structures such as rewired PMA [17] and gapped arrays [20] also touch on this challenge—how to adapt the performance cost of a deterministic LLA algorithm to non-worst case inputs, while still maintaining $O(\log^2 n)$ worst case amortized cost.

A question that looms is whether one can extract improved performance guarantees using learned predictions on general inputs. An ideal algorithm can leverage accurate predictions to give much stronger guarantees than the worst case. Further, the data structure should maintain an $O(\log^2 n)$ guarantee in the worst case even with completely erroneous predictions. Between, it would be ideal to have a natural trade-off between the quality of predictions and the performance guarantees.

## 1.1 Our Contributions

In this paper, we design a new data structure for list labeling which incorporates learned predictions and bound its performance by the error in the predictions. Our theoretical results show that high-quality predictions allow us to break through worst-case lower bounds and achieve asymptotically improved performance. As the error in the prediction grows, the data structure's performance degrades proportionally, which is never worse than the state-of-the-art LLAs. Thus, the algorithm gives improvements with predictions at no asymptotic cost. This is established in both worst-case

---

[1]We use 'online list labeling' and 'list labeling' interchangeably; the problem is trivial in the offline setting.

and stochastic models. Our experiments show that these theoretical improvements are reflected in practice.

**Prediction model.** We define the model to state our results formally. Each element $x$ has a ***rank*** $r_x$ that is equal to the number of elements smaller than it after all elements arrive. When an element arrives, a ***predicted rank*** $\widetilde{r}_x$ is given. The algorithm can use the predicted ranks when storing elements and the challenge is that the predictions may be incorrect. Define $\eta_x = |\widetilde{r}_x - r_x|$ as $x$'s ***prediction error***. Let $\eta = \max_x \eta_x$ be the ***maximum error*** in the predictions.

**Upper bounds.** In Section 3, we present a new data structure, called a ***learnedLLA***, which uses any list labeling array as a black box. We prove that it has the following theoretical guarantees.

- Under arbitrary predictions, the learnedLLA guarantees an amortized cost of $O(\log^2 \eta)$ per insertion when using a classic LLA [30] as a black box. This implies that the algorithm has the best possible $O(1)$ cost when the predictions are perfect (or near perfect). Even when $\eta = n$, the largest possible prediction error, the cost is $O(\log^2 n)$, matching the lower bound on deterministic algorithms. Between, there is graceful degradation in performance that is logarithmic in terms of the error. In fact, these results are more general. Given a list labeling algorithm (deterministic or randomized) with an amortized cost of $C(n)$, for a reasonable[2] function $C$, our data structure guarantees performance $C(\eta)$. This implies the new data structure can improve the performance of any known LLA. See Section 3.2.
- We also analyze the learnedLLA when the error $\eta_x$ for each element $x$ is sampled independently from an *unknown* probability distribution $D$ with average error $\mu$ and variance $s^2$. Under this setting, we show that the amortized cost is $O(\log^2(|\mu| + s^2))$ in expectation. Interestingly, the arrival order of elements can be adversarial and even depend on the sampled errors. This implies that for any distribution with constant mean and variance, we give the best possible $O(1)$ amortized cost. See Section 3.4.

**Lower bound.** A natural next question is whether the learnedLLA is the best possible or if there is an LLA that uses the predictions in a way that achieves a stronger amortized cost. In Section 3.3, we show our algorithm is the best possible: for any $\eta$, *any* deterministic algorithm must have amortized cost at least $\Omega(\log^2 \eta)$, matching our upper bound. Thus, our algorithm utilizes predictions optimally.

**Empirical results.** Finally, we show that the theory is predictive of empirical performance in Section 4. We demonstrate on real datasets that: (1) the learnedLLA outperforms state-of-the-art baselines on numerous instances, (2) when current LLAs perform well, the learnedLLA matches or even improves upon their performing, indicating minimal overhead from using predictions, (3) the learnedLLA is robust to prediction errors, (4) these results hold on real-time-series datasets where the past is used to make predictions for the future (which is typically the use case in practice), and (5) we demonstrate that a small amount of data is needed to construct useful predictions.

## 1.2 Related Work

See [44] for an overview of learning-augmented algorithms in general.

**Learned replacements of data structures.** The seminal work on learning-augmented algorithms by Kraska et al. [35], and several follow up papers [20, 25, 28, 32, 33, 39, 55] focus on designing and analyzing a learned index, which replaces a traditional data structure like a $B$-tree and outperforms it empirically on practical datasets. Interestingly, the use of learned indices directly motivates the learned online list labeling problem. To apply learned indices on dynamic workloads, it is necessary to efficiently maintain the input in sorted order in an array. Prior work [20] attempted to address this through a greedy list labeling structure, a *gapped array*. A gapped array, however, can easily incur $\Omega(n)$ element movements per insert even on non-worst case inputs. This bottleneck does not manifest itself in the theoretical or empirical performance of learned indices so far as the input is assumed to be randomly shuffled. Note that random order inserts are the *best case* for list labeling and incur $O(1)$ amortized cost in expectation [5]. In contrast, our guarantees hold against inputs with adversarial order that can even depend on the prediction errors.

Besides learned indices replacing $B$-trees, other learned replacements of data structures include hash tables [24, 45] and rank-and-select structures [9, 26, 27].

---

[2]Specifically, for admissible cost functions $C(n)$ as defined in Definition 1.

**Learned adaptations of data structures.** In addition to approaches that use a trained neural net to replace a search tree or hash table, several learned variants of data structures which directly adapt to predictions have also been designed. These include learned treaps [12, 38], filters [8, 40, 47, 49] and count-min sketch [22, 29]. The performance bounds of most of these data structures assume perfect predictions; when robustness to noise in predictions is analyzed (see e.g. [38]), the resulting bounds revert to the worst case rather than degrading gracefully with error. The learnedLLA is unique in the landscape of learned data structure as it guarantees (a) optimal bounds for any error, and (b) best worst-case performance when the predictions are entirely erroneous.

**Online list labeling data structures.** The online list labeling problem is described in two different but equivalent ways: storing a dynamic list of $n$ items in a sorted array of size $m$; or assigning labels from $\{1, \ldots, m\}$ to these items where for each item $x$, the label $\ell(x)$ is such that $x < y \implies \ell(x) < \ell(y)$. We focus on the *linear* regime of the problem where $m = cn$, and $c > 1$ is a constant.

If an LLA maintains that any two elements have $\Theta(1)$ empty slots between them, it is referred to as a ***packed-memory array*** (PMAs) [4]. PMAs are used as a subroutine in many algorithms: e.g. applied graph algorithms [16, 18, 42, 48–50] and indexing data structures [4, 20, 34].

A long standing open question about LLAs—whether randomized LLA algorithms can perform better than deterministic was resolved in a recent breakthrough. In particular, Bender et al. [3] extended the *history independent* LLA introduced in [1] and showed that it guarantees $O(\log^{3/2} n)$ amortized expected cost. The learnedLLA achieves $O(\log^{3/2} \boldsymbol{\eta})$ amortized cost when using this LLA as a black box inheriting its strong performance; see Corollary 5.

Bender and Hu [7] gave the first beyond-worst-case LLA, the ***adaptive PMA (APMA)***. The APMA has $O(\log n)$ amortized cost on specialized sequences: *sequential* (increasing/ decreasing), *hammer* (repeatedly inserting the predecessor of an item) and *random* inserts. APMA also guarantees $O(\log^2 n)$ amortized cost in the worst case. Moreover, the APMA uses heuristics to attempt to improve performance on inputs outside of these specialized sequences. These heuristics do not have theoretical guarantees, but nonetheless, the APMA often performs better than a classic one empirically [7, 16]. While the learnedLLA bounds (based on prediction error) are incomparable with that of the APMA, our experiments show that the learnedLLA outperforms the APMA on numerous datasets. In fact, if we combine our techniques by using an adaptive PMA as the black-box LLA of the learnedLLA, it performs better than both of them, reinforcing its power and versatility.

## 2 Preliminaries

In this section we formally define the list labeling problem, our prediction model, and the classic list labeling data structures we use as a building block in Section 3.

**Problem definition.** In the ***online list labeling problem***, the goal is to maintain a dynamic sorted list of $n$ elements in an array of $m = cn$ slots for some constant $c > 1$. We refer to $n$ as the ***capacity*** of the array, $m$ as the ***size***, and $n/m$ as the ***density***. The **label** of an element is the array slot assigned to it. As new elements arrive, existing elements may have to be moved to make room for the new element. All elements come from an ordered universe of possible elements $U$. The list labeling structure maintains that the elements in the array must be stored in sorted order—for any $x_i, x_j$ stored in the array, if $x_i < x_j$ then the label of $x_i$ must be less than the label of $x_j$.

We refer to a data structure for this problem as a ***list labeling array (LLA)***. An ***element movement*** is said to occur each time an element is assigned a new label.[3] We use the term element *movements* and *relabels* interchangeably. The ***total cost*** of an LLA is the total number of element movements. The ***amortized cost*** of an LLA is the total cost during $k$ insertions, divided by $k$. This cost function is standard both in the theory and practice of LLAs (see e.g. [2, 3, 7, 17, 30]).

**Data structure model.** We assume that the set $S$ of $n$ elements are drawn one-by-one adversarially from $U$ and inserted into the LLA. We use existing LLAs (e.g. [1–3, 30]) as a black box in our data structure. We consider any (possibly randomized) LLA $A$ which supports the following operations:

- INSERT($x$): Inserts $x$ in $A$. The slot storing $x$ must be after the slot storing its predecessor, and before the slot storing its successor.

---

[3]This means that every element has at least one element movement when it is inserted.

- INIT($S'$): Given an empty LLA $A$ and given a (possibly-empty) set $S' \subseteq U$, insert all elements from $S'$ in $A$.

Note that INIT($S'$) can be performed using $O(|S'|)$ element movements—as all $|S'|$ elements are available offline, they can be placed in $A$ one by one.

For simplicity we assume that $n$ is a power of 2; our results easily generalize to arbitrary $n$. All logarithms in this paper are base 2.

In this work, we do not explicitly discuss deletes. However, our data structure can be easily extended to support deletes as follows. We build a learnedLLA with capacity $2n$. We split all operations into *epochs* of length $n$. During an epoch we ignore all deletes. After the epoch completes, we rebuild the data structure, removing all elements that were deleted during the epoch. This costs $O(n)$, so the amortized cost is increased by $O(1)$.

We parameterize the amortized cost of the learnedLLA by the amortized cost function of INSERT for a black-box LLA algorithm of density $1/2$.[4] For background on how a classic LLA, the packed-memory array (PMA) [4, 30], works see the full version in the Supplementary Material of the paper. Let $C(n')$ upper bound the amortized cost to insert into a LLA with capacity $n'$ and size $2n'$.

**Definition 1.** *We say that a cost function $C(\cdot)$ is **admissible** if: (1) for all $i, j$ with $i < 2j$, $C(i) = O(C(j))$, (2) $C(n') = \Omega(\log n')$, and (3) for all $j$, $\sum_{i=1}^{\infty} C(2^{j+i})/2^i = O(C(j))$.*

Both $C(n) = \Theta(\log^2 n)$ (the optimal cost function for any deterministic LLA) and $C(n) = \Theta(\log^{3/2} n)$ (the best-known cost function for any randomized LLA [3]) are admissible. There is an unconditional lower bound that any LLA must have amortized cost at least $\Omega(\log n')$ [11].

## 3 Our Data Structure: the learnedLLA

In this section we design and analyze our data structure, the learnedLLA, which uses any generic list labeling array as a black box. Due to space restrictions some proofs are deferred; they can be found in the supplementary materials or the full version of the paper.

In Section 3.1 we describe the learnedLLA. In Section 3.2, we analyze the learnedLLA under arbitrary predictions. In Section 3.3, we show that the learnedLLA is *optimal* for deterministic solutions to list labeling. Finally, in Section 3.4, we bound the performance of the learnedLLA under stochastic predictions (with adversarial insert order).

**Learned list-labeling array.** Our learned list labeling data structure partitions its array into subarrays, each maintained as its own list labeling array. At a high level, when an element $x_i$ is inserted with its predicted rank $\widetilde{r}_i$, we use $\widetilde{r}_i$ to help determine which LLA to insert into while maintaining sorted order. Intuitively, the performance comes from keeping these constituent LLAs small: if the predictions are good, it is easy to maintain small LLAs. But, if some element has a large prediction error, its LLA grows large, making inserts into it more expensive.

### 3.1 Data Structure Description

The learnedLLA with capacity $n$ operates on an array of $m = 6n$ slots. At all times, the learnedLLA is partitioned into $\ell$ ***actual LLAs*** $P_1, P_2, \ldots P_\ell$ for some $\ell$. In particular, we partition the $m$ array slots into $\ell$ contiguous subarrays; each subarray is handled by a black-box LLA. We maintain that for all $i < j$, all elements in $P_i$ are less than all elements in $P_j$.

We define a tree to help keep track of the actual LLAs. Consider an implicit, static and complete binary tree $T$ over ranks $\{1, \ldots n\}$. Each node in $T$ has a set of ***assigned ranks*** and ***assigned slots***. Specifically, the $i$th node at height $h$ has $2^h$ assigned ranks $\{2^h(i-1) + 1, 2^h(i-1) + 2, \ldots, 2^h i\}$, and $6 \cdot 2^h$ assigned slots $\{2^h \cdot 6(i-1) + 1, 2^h \cdot 6(i-1) + 2, \ldots, 2^h \cdot 6i\}$.

We use each actual LLA as a black box to handle how elements are moved within its assigned slots. To obtain the learnedLLA label for an element $x_i$ in actual LLA $P_j$, we sum the black-box label of $x_i$ in $P_j$ and the value of the smallest slot assigned to $P_j$ minus 1.

---

[4]If we initialize a LLA $P_j$ with $m'$ slots, we never store more than $m'/2$ elements in $P_j$. However, this choice is for simplicity, and our results immediately generalize to LLAs using different parameter settings.

Every actual LLA has a set of assigned ranks and assigned slots; these ranks and slots must be from some node in $T$. We say that the LLA **corresponds** to this node in $T$. Each root-to-leaf path in $T$ passes through exactly one node that corresponds to an actual LLA. This means that the assigned ranks of the actual LLAs partition $\{1, \ldots, n\}$ into $\ell$ contiguous subsets, and the assigned slots of the actual LLAs partition $\{1, \ldots, m\}$ into $\ell$ contiguous subsets.

If a node $v$ in $T$ is not assigned to an actual LLA, it is useful to consider the assigned ranks and slots that would be used if a LLA assigned to $v$ were to exist. We call such a LLA a **potential LLA**. Any actual LLA is also a potential LLA. The **sibling**, **parent** and **descendants** of a LLA $P_j$ refer to the LLAs corresponding to the sibling, parent and descendant nodes respectively of $P_j's$ node in the tree.

For any potential LLA $P_j$, let $|P_j|$ denote the number of assigned ranks of $P_j$. Initially $\ell = n$ and $P_i$ is assigned to the $i$th leaf of $T$, where $1 \leq i \leq n$—that is to say, the actual LLAs begin as the leaves of $T$, each with one assigned rank. The parameter $\ell$ changes over time as elements are inserted.

Each element is inserted into exactly one actual LLA. If an LLA has density more than $1/2$, we **merge** it with the (potential or actual) LLA of its sibling node in $T$. This process is described below.

**Insertion algorithm.** To insert an item $x$ into the learnedLLA, let $i_p$ and $i_s$ be the index of the LLA containing the predecessor and successor of $x$ respectively. If the successor of $x$ does not exist, let $i_s = \ell$ (the last actual LLA); if the predecessor does not exist let $i_p = 1$. Finally, let $i_x$ be the LLA whose assigned ranks contain $\widetilde{r_x}$. Call INSERT$(x)$ on LLA $P_i$, where: if $i_p > i_x$ then $i = i_p$; if $i_s < i_x$ then $i = i_s$, and otherwise $i = i_x$.

The black-box LLA $P_i$ updates its labels internally, and these updates are mirrored in the learnedLLA.

If the insert causes the density of $P_i$ to go above $1/2$, we **merge** it with its sibling LLA. Specifically, let $P_p$ be the parent of $P_i$; it must currently be a potential LLA. We take all actual LLAs that are descendants of $P_p$, and merge them into a single actual LLA. Let $S_p$ be the set of elements currently stored in the slots assigned to $P_p$ in the learnedLLA; thus, $S_p$ consists exactly of the contents of all actual LLAs that are descendants of $P_p$. We call INIT$(S_p)$ on $P_p$, after which $P_p$ is an actual LLA; all of its descendants are no longer actual LLAs. Note that after a merge, the assigned ranks and assigned slots of the actual LLAs still partition $\{1, \ldots, n\}$ and $\{1, \ldots, m\}$ respectively.

## 3.2 Arbitrary Predictions

In this section, we analyze the learnedLLA when the input sequence and associated predictions can be arbitrary. The adversary chooses elements one by one from the universe $U$. For each element, the adversary assigns a predicted rank (which can be based on past insertions, past predictions, and even the learnedLLA algorithm being used) and inserts the element. We show that the learnedLLA achieves amortized cost $O(C(\boldsymbol{\eta}))$, where $\boldsymbol{\eta}$ is the maximum error and $C(\cdot)$ is the admissible cost function for the amortized cost of inserting into the black box LLA being used.

Our proof begins with several structural results, from which the analysis follows almost immediately.

First, we show that to determine the overall asymptotic cost it is sufficient to consider the set of actual LLAs after all inserts are completed—relabels during merges are a lower-order term.

**Lemma 2.** *For any sequence of insertions, if $\mathcal{P}_F$ is the set of actual LLAs after all operations are completed, and $C(\cdot)$ is an admissible cost function for the LLAs, then the total number of element movements incurred by the learnedLLA is $O\left(\sum_{P \in \mathcal{P}_F} |P| \cdot C(|P|)\right)$.*

*Proof.* For a LLA $P \in \mathcal{P}_F$, consider all potential LLAs that are descendants of $P$ in $T$. Every $P'$ that is ever an actual LLA must be a descendant of exactly one $P \in \mathcal{P}_F$.

First, we analyze the cost of all merges in descendants of $P$. The total capacity of all LLAs that are a descendant of $P$ is $|P| \log |P|$. Since a merge operation has linear cost, the total number of element movements during all merges is $O(|P| \log |P|)$.

Now, we must bound the cost of all inserts to descendants of $P$. Consider all inserts into all LLAs $P_1, P_2, \ldots, P_d$ that are a descendant of $P$. Let $k_i$ be the number of inserts into some such LLA $P_i$; the total cost of these inserts is $O(k_i C(|P_i|)) \leq O(k_i C(|P|))$. All inserts into descendants of $P$ are ultimately stored in $P$; thus, $\sum_{i=1}^g k_i \leq 3|P|$. Then the total cost of all inserts into LLAs that are a descendant of $P$ is $\sum_{i=1}^g O(k_i C(|P|)) = O(|P|C(|P|))$.

Summing between the merge and insert cost, and using $C(|P|) + \log|P| = O(C(|P|))$ because $C(\cdot)$ is admissible, we obtain a total number of element movements of $O\left(\sum_{P \in \mathcal{P}_F} |P| \cdot C(|P|) + |P|\log|P|\right) \leq O\left(\sum_{P \in \mathcal{P}_F} |P| \cdot C(|P|)\right)$. $\qquad\square$

**Lemma 3.** *If $P$ is an actual LLA, then there exists an element $x_j$ stored in $P$ with $\eta_j \geq |P|/2$.*

*Proof.* Since $P$ was formed by merging its two children $P_i$ and $P_k$ in $T$, the density of either $P_i$ or $P_k$ must have been at least $1/2$; without loss of generality assume it was $P_i$.

Let $r_1, r_1 + 1, \ldots, r_2$ be the sequence of $|P_i|$ ranks assigned to $P_i$. Let $x_s$ and $x_\ell$ be the smallest and largest items in $P_i$ respectively, with predicted ranks $\widetilde{r_s}$ and $\widetilde{r_\ell}$. We must have $\widetilde{r_s} \geq r_1$—since the predecessor of $x_s$ is not in $P_i$, $x_s$ must be placed in the LLA whose assigned ranks contain $\widetilde{r_s}$, or in the LLA containing its successor (if $\widetilde{r_s}$ is larger than any rank assigned to the LLA containing its successor). Similarly, $\widetilde{r_\ell} \leq r_2$. Therefore, $\widetilde{r_\ell} - \widetilde{r_s} \leq |P_i| - 1$. There are at least $3|P_i|$ items in $P_i$ as its density is at least $1/2$. Thus, $r_\ell - r_s \geq 3|P_i| - 1$. Thus, either $|\widetilde{r_s} - r_s| \geq |P_i|$ or $|\widetilde{r_\ell} - r_\ell| \geq |P_i|$. Noting that $|P_i| = |P|/2$, either $\eta_s \geq |P|/2$ or $\eta_\ell \geq |P|/2$. $\qquad\square$

**Theorem 4.** *For any sequence of at most $n$ insertions $\sigma$ with maximum error $\eta$, the learnedLLA using LLAs with admissible cost function $C(\cdot)$ incurs $O(nC(\eta))$ total element movements.*

*Proof.* By Lemma 3, any $P \in \mathcal{P}_F$ must have an element with error at least $|P|/2$. Thus, any $P \in \mathcal{P}_F$ must have $|P| \leq 2\eta$. The actual LLAs partition the ranks, so $\sum_{P \in \mathcal{P}_F} |P| = n$. By Lemma 2, and since $C(\cdot)$ is admissible, the total number of element movements over $\sigma$ is $O\left(\sum_{P \in \mathcal{P}_F} |P| \cdot C(|P|)\right) \leq O\left(\sum_{P \in \mathcal{P}_F} |P| \cdot C(2\eta)\right) \leq O(nC(\eta))$. $\qquad\square$

**Corollary 5.** *Using the classic PMA as an LLA [4, 30], the learnedLLA is deterministic and achieves $O(\log^2 \eta)$ amortized element movements. Using the best-known randomized LLA [3], the learnedLLA achieves $O(\log^{3/2} \eta)$ amortized element movements.*

### 3.3 Optimality for Deterministic Learned List Labeling

The learnedLLA is clearly optimal on the two extremes: perfect predictions ($\eta = 0$) and completely erroneous predictions ($\eta = n$). When predictions are perfect, the learnedLLA does not need to move any element. When $\eta = n$, it is easy to create instances where the predictions give no information about the rank of each item (e.g. if each prediction is $\widetilde{r_i} = 1$). Bulánek et al. [10] showed that, without predictions, any deterministic LLA with $O(n)$ slots can be forced to perform $\Omega(\log^2 n)$ amortized element movements. Thus, the learnedLLA is optimal when $\eta = n$.

However, it is not clear if the above idea extends to intermediate error. In this section, we show that for deterministic LLAs, the learnedLLA is in fact optimal for all $\eta$.

The intuition behind our proof is to split the learned list labeling problem into a sequence of $\Omega(n/\eta)$ subproblems of size $\eta$, each handled by its own LLA algorithm. The given predictions within each subproblem are only accurate to within $\eta$—therefore, an adversary can force each LLA to get *no* information about an element's position within the LLA, and by [10] perform no better than $\Omega(\log^2 \eta)$ amortized element movements.

The challenge is that the LLAs for the subproblems are not actually separate: we can't force the data structure to allocate $O(\eta)$ space to each LLA and keep this allocation static throughout the execution. The data structure may move elements between adjacent LLAs or shift the number of slots available to each, an ability that disrupts a black-box lower bound.

**Lower bound summary.** First, we define a new problem, ***shifted list labeling***, which generalizes online list labeling to allow labels that "shift" over time rather than be confined to $\{1, \ldots, m\}$. We show that this generalization does not increase cost: a data structure for shifted list labeling implies a data structure for online list labeling with the same asymptotic space and amortized cost.

Then, we partition the universe $U$ into $n/\eta$ blocks. The adversary only inserts an element from a block if its elements are spread over at most $O(\eta)$ slots. Thus, the labels of the elements inserted from a particular block form a solution to the shifted list labeling problem—this addresses the challenge of handling separate LLA subproblems. We apply the lower bound of [10] to items inserted from each block; therefore, any block from which $\eta$ items are inserted must have $\Omega(\eta \log^2 \eta)$ amortized cost.

Finally, the adversary implements a cleanup step which inserts extra elements to ensure that the predictions of all elements are accurate to within $\boldsymbol{\eta}$. We show that a constant fraction of the blocks had $\boldsymbol{\eta}$ elements inserted adversarially before the cleanup step. Each of these blocks must have had $\Omega(\boldsymbol{\eta} \log^2 \boldsymbol{\eta})$ element movements; summing we obtain the following theorem.

**Theorem 6.** *For any $n$ and $\boldsymbol{\eta}$, there exists an adversary that inserts $n$ items, each with error at most $\boldsymbol{\eta}$, such that any deterministic data structure for learned list labeling with capacity $n$ and $m = cn$ slots incurs $\Omega(n \log^2 \boldsymbol{\eta})$ total element movements.*

### 3.4 Stochastic Predictions with Adversarial Insert Order

In this section, we assume that each predicted rank is the result of adding random noise to the true rank. In particular, given an *unknown* distribution $\mathcal{D}$, for any item $x_i$, we have $\widetilde{r}_i = r_i + e_i$ where each $e_i$ is sampled independently from $\mathcal{D}$. This means that the error of each item is essentially sampled from $\mathcal{D}$, as $\boldsymbol{\eta}_i = |e_i|$. The arrival order of the elements can still be chosen adversarially, with full knowledge of the predicted ranks. In particular, we can describe our input model with the following adversary. The adversary first chooses $n$ elements of the set $S$ from the universe $U$. For each element $x_i \in S$, the error $e_i$ is then sampled from $\mathcal{D}$. Finally, the adversary can look at the predicted ranks and choose any insert order of elements in $S$. In this model, we prove the following.

**Theorem 7.** *Consider a learnedLLA data structure using a black box LLA algorithm with admissible cost $C(\cdot)$. The total element movements incurred over a sequence of at most $n$ insertions with stochastic predictions is $O\left(n \cdot C((|\mu| + s^2)^2)\right)$.*

As in Corollary 5, we immediately obtain a deterministic learnedLLA with $O(\log^2(|\mu|+s))$ expected amortized cost, and a randomized learnedLLA with $O(\log^{3/2}(|\mu| + s))$ expected amortized cost.

## 4 Experiments

This section presents experimental results on real data sets. The goal is to show that the theory is predictive of practice. In particular, the aim is to establish the following.

- The learnedLLA improves performance over baseline data structures. Moreover, for the common use case of temporal data, learning from the past leads to future improvements.
- Predictions made on only a small amount of past data lead to improved performance.
- The learnedLLA is robust to large errors in the predictions.

**Experimental setup.** We compare the performance of two strong baselines with our algorithm. The first baseline is the Packed-Memory Array (PMA) [4] and the second is the Adaptive Packed-Memory Array (APMA) [7]. We tested LearnedLLA with both of these LLAs as the black box LLA; we call these the LearnedLLA + PMA and LearnedLLA + APMA. To ensure all algorithms have the same memory, we implemented the PMA (resp. APMA) as a LearnedLLA using a PMA (resp. APMA) as a black box, placing each element into the first black box LLA.

We use datasets from SNAP Large Network Dataset Collection [37]. All the datasets we use are temporal. The timestamp is used for the arrival order, and an element feature is used for the value.

To generate the predictions for LearnedLLA, we use a contiguous subsequence $L_{\text{train}}$ of the input in temporal order as our training data. Our test data $L_{\text{test}}$ is a contiguous subsequence of the input that comes right after $L_{\text{train}}$, again in temporal order.

We use two different algorithms for obtaining predictions:

- $\text{predictor}_1(L_{\text{train}}, L_{\text{test}})$: For each element $x \in L_{\text{test}}$, this function first finds the rank of $x$ in $L_{\text{train}}$, and then it scales it by $|L_{\text{test}}|/|L_{\text{train}}|$. Finally, it returns these predictions.
- $\text{predictor}_2(L_{\text{train}}, L_{\text{test}})$: Let $a$ be the slope of the best-fit line for points $\{(i, L_{\text{train}}[i])\}_{1 \le i \le |L_{\text{train}}|}$. First, this function adds $a \cdot (d + i \cdot (\frac{|L_{\text{test}}|}{|L_{\text{train}}|} - 1))$ to the $i$'th element in $L_{\text{train}}$, for each $1 \le i \le |L_{\text{train}}|$, to obtain $L'_{\text{train}}$, where $d$ is the difference between the starting points of training and test data in our input sequence. Then it returns $\text{predictor}_1(L'_{\text{train}}, L_{\text{test}})$.

Let $L^1_{\text{train}}$ and $L^2_{\text{train}}$ be the first and second halves of $L_{\text{train}}$, respectively. To obtain the final predictions for LearnedLLA, we calculate the predictions $P_i := \text{predictor}_i(L^1_{\text{train}}, L^2_{\text{train}})$ for $i = 1, 2$,

and run LearnedLLA on $L_{\text{train}}^2$ with both of these predictions. If $P_{i*}$ performs better, we use $\text{predictor}_{i*}(L_{\text{train}}, L_{\text{test}})$ as the final predictions for $L_{\text{test}}$.

**Experimental results.** See Table 1 for performance on several real data sets. Due to limited space, we only include plots on one data set: Gowalla [14], a location-based social networking website where users share their locations by checking in. The latitudes of each user are used as elements in the input sequence. The remaining plots are given in the supplmentary materials and the full version of the paper.

|  | Gowalla (LocationID) | Gowalla (Latitude) | MOOC | AskUbuntu | email-Eu-core |
|---|---|---|---|---|---|
| PMA | 7.14 | 14.56 | 19.22 | 24.56 | 21.49 |
| APMA | 7.38 | 15.63 | 16.70 | 10.84 | 21.43 |
| LearnedLLA + PMA | **3.36** | **6.06** | **11.99** | 14.27 | **16.55** |
| LearnedLLA + APMA | **3.36** | 6.15 | 12.13 | **8.49** | **16.55** |

Table 1: Amortized cost of LLAs on several real datasets. In all cases, we use the first and second $2^{17} = 131072$ entries as training and test data, respectively.

Figure 1a shows the amortized cost versus the test data size. For several values for $k$, we use the first and second $n = 2^k$ portions of the input as training data and test data, respectively. Figure 1b shows the performance versus the training data size to illustrate how long it takes for the LearnedLLA to learn. The x-axis is the ratio of the training data size to the test data size (in percentage). Figure 1c is a robustness experiment showing performance versus the noise added to predictions, using half of the data as training. In this experiment, we first generate predictions by the algorithm described above, and then we sample $t$ percent of the predictions uniformly at random and make their error as large as possible (the predicted rank is modified to 1 or $n$, whichever is farthest from the current calculated rank). We repeat the experiment five times, each time resampling the dataset, and report the mean and standard deviation of the results of these experiments.

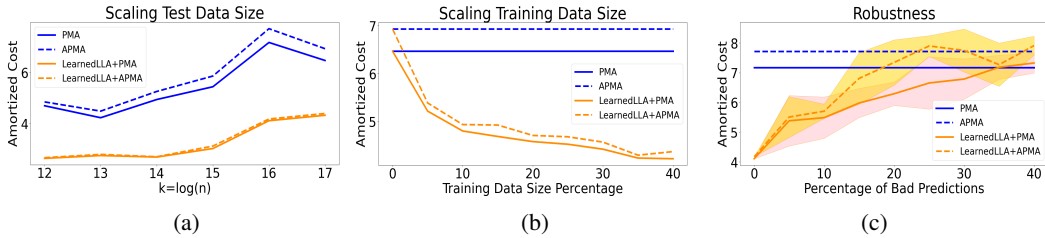

(a)        (b)        (c)

Figure 1: Latitudes in the Gowalla dataset. In Fig. 1a, we use the first and second $n = 2^k$ entries as training data and test data (resp.) for several values of $k$. In Fig. 1b, the test data has a fixed size $n = 2^{17}$ elements, and we increase the size of the training data. In Fig. 1c, training data and test data have size $2^{16}$, and we increase the percentage of "bad" predictions. The lines are the means of the 5 experiments, and the clouds around them show the standard deviation.

**Discussion.** Results in Table 1 show that in most cases, using even a simple algorithm to predict ranks can lead to significant improvements in the performance over the baselines; in some cases by over 50%. Whenever the APMA outperforms the PMA, the learnedLLA inherits this performance gain when using the APMA as a black box, showing its versatility. Furthermore, as illustrated in Figure 1b, a small amount of past data—in some cases as small as 5%—is needed to see a significant separation between the performance of our method and the baseline LLAs. Finally, Figure 1c suggests that our algorithm is robust to bad predictions. In particular, in this experiment, the maximum error is as large as possible and a significant fraction of the input have errors in their ranks, yet the LearnedLLA is still able to improve over baseline performance. We remark that when an enormous number of predictions are completely erroneous, the method can have performance worse than baselines.

In the supplementary materials and the full version of the paper, we include more details about the above experiments and datasets, as well as more experiments which further support our conclusions.

# 5 Conclusion

In this paper, we show how to use learned predictions to build a learned list labeling array. We analyze our data structure using the learning-augmented algorithms model. This is the first application of the model to bound the theoretical performance of a data structure. We show that the new data structure optimally makes use of the predictions. Moreover, our experiments establish that the theory is predictive of practical performance on real data.

An exciting line of work is to determine what other data structures can have improved theoretical performance using predictions. A feature of the list labeling problem that makes it amenable to the learning-augmented algorithms model is that its cost function and online nature is similar to the competitive analysis model, where predictions have been applied successfully to many problems. Other data structure problems with similar structure are natural candidates to consider.

## Acknowledgments and Disclosure of Funding

Samuel McCauley was supported in part by NSF CCF 2103813. Benjamin Moseley was supported in part by a Google Research Award, an Infor Research Award, a Carnegie Bosch Junior Faculty Chair, National Science Foundation grants CCF-2121744 and CCF-1845146 and U. S. Office of Naval Research grant N00014-22-1-2702. Aidin Niaparast was supported in part by U. S. Office of Naval Research under award number N00014-21-1-2243 and the Air Force Office of Scientific Research under award number FA9550-20-1-0080. Shikha Singh was supported in part by NSF CCF 1947789.

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
