# OpenReview forum: "Online List Labeling with Predictions"
_NeurIPS.cc/2023/Conference — NeurIPS 2023 spotlight_

### Official Review · Reviewer_v8Wd · 2023-07-02

**Soundness:** 4 excellent
**Presentation:** 4 excellent
**Contribution:** 3 good
**Rating:** 8
**Confidence:** 4

**Summary:**

This paper is about the classic Online List Labeling problem in the recent model of algorithms with predictions (a.k.a. learning-augmented algorithms).

In this problem n orderable items (say, integers) arrive over time. We need to keep them in the sorted order in an array of size c*n, for some constant c. When the i-th item arrives, we know its relative ordering only with respect to the i-1 items that arrived before, so we do not know where to put it, and we might need to reinsert it later at a different location (in order to have space for future items while maintaining the sorted order). Our goal is to minimize the number of such reinsertions.

There is already known a deterministic data structure with amortized cost O(log^2 n) per insert, and a randomized one with cost O(log^3/2 n).

In this paper the authors propose a data structure that together with each new item receives a prediction on what would be the final rank of that item among all n items.

The data structure uses internally as a black-box a classic (prediction-less) data structure for the problem. Denoting by f(n) the cost of this internal black-box data structure – so, f(n) = O(log^2 n) if we want a deterministic solution, and f(n) = O(log^3/2 n) if we are fine with a randomized one – and assuming all predicted ranks are within eta from the true ranks, the amortized cost of the new algorithm is O(f(eta)). Moreover, if prediction errors are distributed according to a Gaussian distribution with mean mu and variance var, the cost is O(f((mu + var)^2)).

The main idea of the data structure is to maintain a tree-like structure over the inserted items, and delegate maintaining items in non-overlapping subtrees of different sizes to instances of a classic data structure for the problem. Locations of items in the tree are guided by their predicted ranks. If a data structure corresponding to a subtree becomes overfull because of prediction errors, it gets merged with neighboring subtrees. This general idea was already present in the literature around the problem, but the details of implementation and analysis are very novel.

The authors also provide results of some simple experiments. What is nice (and always present in learning-augmented literature) is that the predictions are actually learned from training data, so the experiments can be considered end-to-end. The cost improvements observed over classic data structures are roughly 50%.

**Strengths:**

1) The main algorithm is nontrivial, yet simple and with a clearly visible main idea.

2) I like the additional analysis of the algorithm's performance under prediction errors coming from a Gaussian distribution (on top of the more standard analysis in terms of the maximum prediction error).

3) The algorithm uses existing classic data structures in a black-box manner, so it should be easy to implement and can leverage already optimized implementations of these data structures.

4) The authors provide a matching lower bound (albeit only for deterministic algorithms).

5) The paper is well written and easy to follow.

Simply speaking, it is one of the most interesting recent papers that I have read in this area.

**Weaknesses:**

1) The studied problem seems relevant in practice, but I am not sure how important it really is – however, even if it is not, I find the paper sufficiently appealing on purely theoretical grounds to merit acceptance.

2) The experiments are of a proof-of-concept style – e.g., datasets are adapted from benchmarks for other problems that have not much to do with this one – but it is a common thing for papers in this area.

**Questions:**

Would it be feasible to prove bounds in terms of average L1 prediction error per item, instead of L_inf error? In other words, is it really the case that only a few items with terrible predictions can ruin performance of the whole data structure?

I'd be very interested in seeing results of similar experiments but done in an online learning, rather than batch learning, fashion. That is, using all the data as test data, and predicting rank of item i based on its relative rank among the previous i-1 items (possibly with the linear interpolation trick that you employed and some learning-with-experts algorithm telling whether to use the trick or not).

The claim in line 37 (that there are no earlier works that analyze effect of errors on learning-augmented data structures) seems not very accurate – see, e.g., the section on robustness guarantees in [12]. Similarly, a similar claim in lines 403–404 seems unjustified.

Minor remarks:

Line 54: "moves O(log^2 n) elements" -> "moves O(log^2 n) elements per insert"

Line 56: "a \Omega" -> "an \Omega"

Line 67: "a LLA" -> "an LLA"

Line 68: "worst case" – if the bounds are only amortized, this is misleading

Lines 146–150: you mention the randomized result very late, I think it should be mentioned already in line 57, where you talk about the deterministic result.

Line 72: "would ideal" -> "would be ideal"

Line 96: the footnote mark's location is unfortunate, it reads "C squared"

Line 194: "(PMA)[4,30]" -> "(PMA) [4,30]"

Line 220: it would be nice to mention that such idea of a tree-like data structure already appears in the classic PMA

Line 243: "i_p=\ell" -> "i_s=\ell"

Line 329: "that inserts n items" -> "that, for any n, inserts n items"

Line 515 in the full version: "lemma" -> "theorem"

**Limitations:**

There are only some minor limitations to this work:

1) The lower bound holds only against deterministic algorithms.

2) The performance guarantees are using the L_infty error measure, so it is not clear how the algorithm behaves when there is one outlier item with huge error but all other items are predicted almost correctly. The natural distribution analysis gives some hints that the performance of the algorithm should be good in such a case, but there is no formal proof.

3) There are only pretty basic proof-of-concept experiments.

---

> ### Author Rebuttal · Authors · 2023-08-07
>
> Thank you for the thoughtful review.   We greatly appreciate your comments on writing and will incorporate them in the final version of the paper.
>
> **Question:** would it be feasible to prove bounds in terms of average L1 prediction error per item, instead of L_inf error? In other words, is it really the case that only a few items with terrible predictions can ruin performance of the whole data structure?
>
> **Answer:** The paper introduces a stochastic model to show the algorithm has strong performance when the average error and variance are small.  We elaborate more on this below. Furthermore, we show the algorithm is robust empirically to the case where even a constant fraction of the elements have the maximum possible error (so long as a constant fraction of the elements have reasonably good predictions).
>
> *Stochastic Model:* We assume each element has an error drawn from a probability distribution.  The distribution can be arbitrary.  We show that the amortized cost is $O(\log(\mu s^2))$ where $\mu$ is the mean of the distribution and $s^2$ is the variance.  Intuitively, so long as the variance and mean error are reasonable, this guarantees much better than worst-case bounds.
>
> *Example in the Stochastic Model*: Say each elements error is drawn from a distribution where with probability $1-1/n^{1/2}$ there error is constant and with probability $1/\sqrt{n}$ the error is huge, say $n^{1/4}$.  In this case, we expect a significant number of elements, $\sqrt{n}$ to have a large error $n^{1/4}$.   However, the mean and variance are $O(1)$ and our algorithm guarantees the best possible (up to constant factors) $O(1)$ amortized look-up.  This works even with a worst-case insert order of these elements.
>
> A new and interesting direction is to consider a model where the average error is small *and* the error over the elements is chosen in a worst-case adversarial sense.  We do not know an algorithm with similar guarantees and we believe addressing this case will require new techniques. We leave it as an intriguing open question.
>
> **Question:** I'd be very interested in seeing results of similar experiments but done in an online learning, rather than batch learning, fashion. That is, using all the data as test data, and predicting rank of item i based on its relative rank among the previous i-1 items (possibly with the linear interpolation trick that you employed and some learning-with-experts algorithm telling whether to use the trick or not).
>
> **Answer:**  We did not consider this case.  This is an interesting direction and we appreciate this comment.  We will investigate this for the journal version of this paper.
>
>
> **Question:** The claim in line 37 (that there are no earlier works that analyze effect of errors on learning-augmented data structures) seems not very accurate – see, e.g., the section on robustness guarantees in [12]. Similarly, a similar claim in lines 403–404 seems unjustified.
>
> **Answer:**  This is a good point. We will rephrase and include a more detailed comparison to guarantees in prior work.

---

> > ### Comment · Reviewer_v8Wd · 2023-08-11
> >
> > Thank you for your careful response. I'll keep my score, it seems to be a clear accept.

---

> > > ### Comment · Reviewer_v8Wd · 2023-08-15
> > >
> > > After calibrating my scores with other papers, I'll increase the initial evaluation to 8.

---

### Official Review · Reviewer_QPB8 · 2023-07-06

**Soundness:** 4 excellent
**Presentation:** 3 good
**Contribution:** 3 good
**Rating:** 8
**Confidence:** 3

**Summary:**

The author presented a novel learning-augmented algorithm for the online list labeling problem, providing solid theoretical guarantees in terms of the error in the predictions. They also investigated the stochastic error model and bound the performance in terms of the expectation and variance of the error. Finally, they carried out an experimental evaluation of the proposed methods.


**Strengths:**

Although the problem studied, online list labeling, might be considered quite peculiar for this conference, I believe it can be a seminal work for learning augmented data structures. I like this work very much because they provide solid performance guarantees, showing that our algorithm utilizes predictions optimally because their cost upper bound matches a lower bound shown in Section 3.3 for any prediction error. They also show that their algorithm is optimal in the case of entirely erroneous predictions and that for deterministic LLAs, the learned LLA is optimal for all prediction errors.
I also appreciated the study of the case in which the error for each element x is sampled independently from an unknown probability distribution with a given average and variance error.

**Weaknesses:**

The main weakness of this work is that the problem studied, online list labeling, is quite specific. It would be interesting to apply this methodology to other data structures to solve problems that are more common and known.

**Questions:**

What other data structures will likely benefit from an approach similar to the one you used in this work?

**Limitations:**

I cannot identify any restrictions or potential negative effects this work may have on society.

---

> ### Author Rebuttal · Authors · 2023-08-07
>
> Thank you for the thoughtful review.
>
> **Question:** What other data structures will likely benefit from an approach similar to the one you used in this work?
>
> **Answer:**  We believe that the model and general algorithmic ideas will be useful for some other data structures.  We have initial results that binary search trees, heaps, and nearest neighbor indices benefit from (1) adapting the learning-augmented framework used in this paper and (2) leveraging the algorithmic idea of partitioning the input and assigning them to black-box data structures based on their predictions.

---

> > ### Comment · Reviewer_QPB8 · 2023-08-15
> > **Other data structures**
> >
> > Thank you for clarifying the point.

---

### Official Review · Reviewer_p1qT · 2023-07-06

**Soundness:** 3 good
**Presentation:** 3 good
**Contribution:** 3 good
**Rating:** 7
**Confidence:** 4

**Summary:**

Author study a fundamental online problem which is also important
in practice: online list labeling.
The introduce rank predictions for this problem and define prediction error
in a natural way. They achieve O(1) consistency and a robustness comparable
with the best possible classical online algorithm.
They show that the dependence of their competitive ratio on prediction error
is the best possible. They also analyze the case where the prediction
error comes from some unknown distribution.
Given the importance of the problem and soundness of the results, I recommend
this paper to be accepted.


**Strengths:**

* important problem
* tight bounds, complete analysis

**Weaknesses:**

* consistency is O(1) but not really close to 1.

**Questions:**



**Limitations:**

* authors state their theoretical results formally, describing all assumptions.

---

> ### Author Rebuttal · Authors · 2023-08-07
>
> Thank you for the thoughtful review.
>
> **Question:** consistency is $O(1)$ but not really close to $1$.
>
> **Answer:** If the predictions are perfect, then every element is inserted and never moves. This gives consistency $1$.  We will clarify this in the paper.

---

> > ### Comment · Reviewer_p1qT · 2023-08-12
> >
> > thank you for the clarification.

---

### Official Review · Reviewer_x3tY · 2023-07-06

**Soundness:** 3 good
**Presentation:** 2 fair
**Contribution:** 2 fair
**Rating:** 6
**Confidence:** 4

**Summary:**

The paper considers the online list labelling problem: A set of $n$ elements arrive online and have to be inserted into an $c\cdot n, c>1$ constant, large array while the array must at all times be sorted. Every time an already inserted element is "shifted" in order to maintain the order, it incurs a cost of $1$ and the goal is to minimize the total cost. The variant of the problem studied here, is the one where alongside the arrival of each element the algorithm obtains a possibly erroneous prediction on the rank of that element.

The paper presents an algorithm, that if the total error $\eta$ is defined as the maximum error in the predictions, has an amortised cost of $O(\log^2 \eta)$ and thus only constant cost when $\eta$ is almost zero. This bound is tight for any deterministic algorithm. These results are complemented with empirical evaluation.

**Strengths:**

It is an interesting problem and data structures with predictions are indeed not as developed as online problems with predictions.

**Weaknesses:**

- The writing is at places not very precise and can be misleading. To give an example saying that the data structure is "optimal for any prediction error" is not true. For one I think the authors use "optimal" to mean "best possible" which differs from the convention that optimal refers to the offline optimal algorithm which knows the whole input in advance. Furthermore even "best possible" would not be completely accurate since it  can be up to a constant factor worse than the best previously known result with predictions. Perhaps "tight up to a constant factor" would be accurate? See also L80 "improvement at no cost", or L79 where "proportionally" I think is meant to be "linearly" etc.

- The prediction error is not very natural: Consider the scenario where one prediction is off by a lot and all others are perfect. Considering the maximum error in the predictions over all elements is heavily unfair against such a scenario where the predictions are arguably quite good and one could easily obtain an optimal insertion of all elements without (or with barely) any shifting.

**Questions:**

- You repeatedly use the term array slot. Although I understand what you mean, this is to my knowledge not common terminology, right?

- How does your algorithm perform if one defines the error to be the average error in the individual predictions? Are there any lower bounds for *any* algorithm in that case?

- L217: what is an "actual LLA"?

**Limitations:**

No limitations or potential negative societal impact anticipated.

---

> ### Author Rebuttal · Authors · 2023-08-07
>
> We thank the reviewer for the helpful comments.  We address your questions below.
>
>
> **Question:**  saying that the data structure is "optimal for any prediction error" is not true. For one I think the authors use "optimal" to mean "best possible" which differs from the convention that optimal refers to the offline optimal algorithm which knows the whole input in advance.
>
> **Answer:**  Indeed, we meant best possible for online algorithms up to constant factors.  We will change the wording to reflect this.
>
> **Question:** The prediction error is not very natural: Consider the scenario where one prediction is off by a lot and all others are perfect. Considering the maximum error in the predictions over all elements is heavily unfair against such a scenario where the predictions are arguably quite good and one could easily obtain an optimal insertion of all elements without (or with barely) any shifting.
>
> **Answer:** It is a fair observation that maximum error may be pessimistic for cases where a single element has large prediction error.   Due to this, we also analyze performance using a stochastic model and show that the algorithm has strong performance when the average error and variance is small.  We elaborate more on this below. Furthermore, we show the algorithm is robust empirically when even a constant fraction of the elements have the maximum possible error (so long as a constant fraction of the elements have reasonably good predictions).
>
> *Stochastic Model:* We assume each element has an error drawn from a probability distribution.  The distribution can be arbitrary.  We show that the amortized cost is $O(\log(\mu s^2))$ where $\mu$ is the mean of the distribution and $s^2$ is the variance.  Intuitively, so long as the variance and mean error are reasonable, this guarantees much better than worst-case bounds.
>
> *Example in the Stochastic Model:* Say each elements error is drawn from a distribution where with probability $1-1/n^{1/2}$ there error is constant and with probability $1/\sqrt{n}$ the error is huge, say $n^{1/4}$.  In this case, we expect a significant number of elements, $\sqrt{n}$ to have large error $n^{1/4}$.   However, the mean and variance are $O(1)$ and our algorithm guarantees the best possible (up to constant factors) $O(1)$ amortized look-up.  This works even with a worst-case insert order of these elements.
>
> **Question:** You repeatedly use the term array slot. Although I understand what you mean, this is to my knowledge not common terminology, right?
>
> **Answer:** We will define this in the preliminaries as a position (i.e. index) in the array.
>
>
> **Question:** How does your algorithm perform if one defines the error to be the average error in the individual predictions? Are there any lower bounds for any algorithm in that case?
>
> **Answer:**  In the paper we give some bounds for this case in the stochastic error model (see above).  Another possibility is to consider a model where the average error is small *and* the error over the elements is chosen in a worst-case adversarial sense.  We do not know an algorithm with similar guarantees and we believe addressing this case will require new techniques. We leave it as an intriguing open question.

---

> > ### Comment · Reviewer_x3tY · 2023-08-11
> >
> > Thanks a lot for the extensive response.
> >
> > After going over the other reviews and the respective responses carefully, I decided to slightly raise my score.

---

### Decision · Program_Chairs · 2023-09-21

**Decision:**

Accept (spotlight)

**Comment:**

This paper studies the online list labeling problem in the learning-augmented setting where the algorithm is given a prediction about the rank of each element upon arrival. The results are very strong, this paper has the potential to have an important impact in the area of learning-augmented data structures. The results are tight for deterministic algorithms and nicely solve the problem studied. I recommended acceptance.